# Methanethiol: A Scent Mark of Dysregulated Sulfur Metabolism in Cancer

**DOI:** 10.3390/antiox12091780

**Published:** 2023-09-19

**Authors:** Thilo Magnus Philipp, Anne Sophie Scheller, Niklas Krafczyk, Lars-Oliver Klotz, Holger Steinbrenner

**Affiliations:** Institute of Nutritional Sciences, Nutrigenomics Section, Friedrich Schiller University Jena, D-07743 Jena, Germany; thilo.magnus.philipp@uni-jena.de (T.M.P.); anne.sophie.scheller@uni-jena.de (A.S.S.); niklas.krafczyk@uni-jena.de (N.K.); lars-oliver.klotz@uni-jena.de (L.-O.K.)

**Keywords:** tumor, CRC, VSC, H_2_S, methyl mercaptan, SELENBP1, MTO, microbiota

## Abstract

In order to cope with increased demands for energy and metabolites as well as to enhance stress resilience, tumor cells develop various metabolic adaptations, representing a hallmark of cancer. In this regard, the dysregulation of sulfur metabolism that may result in elevated levels of volatile sulfur compounds (VSCs) in body fluids, breath, and/or excretions of cancer patients has recently gained attention. Besides hydrogen sulfide (H_2_S), methanethiol is the predominant cancer-associated VSC and has been proposed as a promising biomarker for non-invasive cancer diagnosis. Gut bacteria are the major exogenous source of exposure to this foul-smelling toxic gas, with methanethiol-producing strains such as *Fusobacterium nucleatum* highly abundant in the gut microbiome of colorectal carcinoma (CRC) patients. Physiologically, methanethiol becomes rapidly degraded through the methanethiol oxidase (MTO) activity of selenium-binding protein 1 (SELENBP1). However, SELENBP1, which is considered a tumor suppressor, is often downregulated in tumor tissues, and this has been epidemiologically linked to poor clinical outcomes. In addition to impaired removal, an increase in methanethiol levels may derive from non-enzymatic reactions, such as a Maillard reaction between glucose and methionine, two metabolites enriched in cancer cells. High methionine concentrations in cancer cells may also result in enzymatic methanethiol production in mitochondria. Moreover, enzymatic endogenous methanethiol production may occur through methyltransferase-like protein 7B (METTL7B), which is present at elevated levels in some cancers, including CRC and hepatocellular carcinoma (HCC). In conclusion, methanethiol contributes to the scent of cancer as part of the cancer-associated signature combination of volatile organic compounds (VOCs) that are increasingly being exploited for non-invasive early cancer diagnosis.

## 1. Introduction

In 2000, Hanahan and Weinberg presented their groundbreaking concept of “the hallmarks of cancer”, when they defined six pathophysiological alterations in cancer cells that develop during carcinogenesis and distinguish transformed from non-transformed cells [1]. Later on, the original catalog was extended by two more cancer hallmarks, including the reprogramming of (energy) metabolism [2]. However, the exploration of cancer metabolism has a much longer history: the first observation of altered carbohydrate metabolism in many cancer cells, as characterized by a shift in the predominant pathway of ATP production from oxidative phosphorylation in the respiratory chain to aerobic glycolysis, dates back to landmark studies by Otto Warburg in the 1920s [3,4]. This so-called “Warburg effect” is now interpreted as part of extensive metabolic adaptations that enable cancer cells to generate energy, metabolites, and signaling messengers for growth, progression, and metastasis, as well as to sustain their redox homeostasis [5,6].

In order to feed anabolic growth-promoting pathways, cancer cells may capture multiple nutrients besides glucose, including lactate and the amino acids glutamine, serine, and methionine [5,6,7]. In this regard, cancer cells have recently been discussed to be “addicted” to methionine [7]. The selective upregulation of the methionine transporter SLC43A2 allows cancer cells to outcompete cells in the tumor microenvironment for methionine supply [8]. The essential amino acid methionine is used for the biosynthesis of proteins, nucleotides, glutathione, and the methyl donor S-adenosylmethionine (SAM), which are required for cellular growth and proliferation, as well as protection against oxidative stress [7]. Moreover, methionine serves as a precursor for the endogenous and exogenous generation of two volatile sulfur compounds (VSCs): hydrogen sulfide (H_2_S) and methanethiol. Several strains of gut bacteria may convert dietary methionine to H_2_S and/or methanethiol, thus representing the major source of exposure of humans to both VSCs. In addition, H_2_S and methanethiol are produced endogenously through enzymatic and non-enzymatic synthesis [9,10]. The cancer-associated dysregulation of sulfur metabolism results in excess levels of both VSCs in tumor tissue as well as in body fluids, breath, and/or excretions of cancer patients that are increasingly being exploited for the establishment of convenient non-invasive screening methods to detect early signs of cancer [11,12,13,14]. In this regard, the dysregulated H_2_S biosynthesis in various types of cancer and the cancer-promoting effects of elevated H_2_S levels have recently been discussed in several reviews [10,15,16]. Excess H_2_S may promote cellular dedifferentiation and the growth of transformed cells as well as increase their potential to metastasize and develop resistance against chemotherapeutic agents [10]. The endogenous production of H_2_S in mammalian cells occurs largely through four enzymes: cystathionine β-synthase (CBS), cystathionine γ-lyase (CTH), 3-mercaptopyruvate sulfurtransferase (MPST), and selenium-binding protein 1 (SELENBP1) [9,17,18]. In particular, CBS has been shown to be upregulated in various types of cancer, including colorectal carcinoma (CRC), squamous cell carcinoma, and ovarian, breast, and thyroid cancers [10]. In several of those cancers, CTH and MPST levels were also found to be elevated [10]. On the other hand, SELENBP1, which generates H_2_S through the oxidation of methanethiol [18], is often downregulated in tumor tissue [19].

Methanethiol (methyl mercaptan, CH_3_SH) is an alkyl thiol and a member of a group of repulsive-smelling VSCs, characterized by their toxicity and low odor threshold. It is a colorless gas with a boiling point of 5.9 °C. The smell of methanethiol is reminiscent of rotten cabbage and is perceptible to humans at concentrations of ~1–2 parts per billion (ppb) [20]. A study on rats that investigated the toxicity of different VSCs reported a 24 h LD_50_ value of 675 parts per million (ppm) for acute exposure to methanethiol, which was similar to the LD_50_ of 444 ppm measured for H_2_S [21]. The toxicity of methanethiol has been attributed to its inhibiting effects on cytochrome *c* oxidase and the electron transfer in the respiratory chain [22]. Environmental microbial production of methanethiol occurs mainly through the methylation of H_2_S in the anoxic sediment/water interphase and through the elimination reactions of sulfur-containing amino acids catalyzed by L-methionine-γ-lyase (MGL). In addition, it can be generated from dimethylsulfoniopropionate (DMSP), an osmolyte in marine algae, through a coupled series of bacterial demethylation and cleavage reactions [20]. Although methanethiol has long been known as an environmental toxin and an intermediate in the biogeochemical sulfur cycle, its metabolism in humans has only recently gained more attention, following the identification of human selenium-binding protein 1 (SELENBP1) as a novel methanethiol-oxidizing enzyme in 2018 [18].

In this review, we provide an overview of the physiologically occurring formation and degradation of the VSC methanethiol in humans, as well as the dysregulation of methanethiol metabolism in cancer. The exudation of excess methanethiol and its methylated volatile metabolites by cancer patients may contribute to the “scent of cancer”; thus, we discuss the potential of methanethiol as a biomarker for non-invasive early cancer detection.

## 2. Exogenous and Endogenous Methanethiol Production in Humans

Methanethiol is a common volatile component of the human flatus: in healthy volunteers to whom beans and lactulose were given to enhance their flatus output, the measured methanethiol concentrations reached 0.19–0.24 µmol/L (9.14–11.55 ppb). Thus, among the detected VSCs, methanethiol showed the second highest concentrations after H_2_S [23]. Analogous to H_2_S, the majority of methanethiol biosynthesis in humans likely originates from the gut microbiome. Commensal bacteria residing in the colonic lumen can convert dietary methionine to methanethiol through MGL-catalyzed α,γ-elimination and γ-replacement reactions. *Fusobacterium nucleatum*, *Citrobacter freundii*, *Morganella morganii*, and several *Proteus* species have been reported to contribute to intestinal methanethiol production [24,25]. Individual variations in the composition of the bacterial strains in the colon as well as differences between human populations may, thus, affect the exogenous production of methanethiol. The major role of gut microbiota as methanethiol producers is supported by the demonstration of decreased methanethiol levels in patients who were treated with antibiotics [18]. Methanethiol concentrations released by the intestinal flora are dependent on nutritional factors and may be influenced by the type of diet and dietary restrictions [23]. In particular, the ingestion of dietary proteins that contain high amounts of methionine may result in increased methanethiol production by the gut microbiome, whereas activated charcoal and zinc salts have been demonstrated to neutralize sulfur-containing malodorous gases such as methanethiol and H_2_S [23]. Besides being excreted through flatus and feces, some of the bacteria-derived methanethiol diffuses into the epithelial cells lining the colonic lumen of the human host [9].

Regarding endogenous production, there is experimental evidence for the occurrence of two methanethiol-generating enzymatic pathways in human cells. Recently, recombinant methyltransferase-like protein 7B (METTL7B; *aka* thiol S-methyltransferase TMT1B) has been demonstrated to be capable of producing methanethiol through the methylation of H_2_S, using SAM as the methyl donor [26]. METTL7B is highly abundant in the gut, the liver, the kidneys, and the lungs (https://www.proteinatlas.org/ENSG00000170439-METTL7B/tissue, accessed on 3 July 2023); nevertheless, the quantitative contribution and the physiological relevance of this novel methanethiol-generating enzymatic pathway remain to be determined. Furthermore, a study on isolated mitochondria found that methanethiol may arise from the oxidation of the transamination product of methionine, α-keto-methylthiobutyrate [27]. However, this pathway might be quantitatively relevant only under conditions of methionine excess. In this regard, mice fed a methionine-rich diet showed characteristic features of methanethiol toxicity [28]. Polymorphisms in the genes, coding for the enzymes that are responsible for endogenous methanethiol production, may also have an influence on the cellular and systemic methanethiol concentrations; however, this remains to be explored yet.

## 3. SELENBP1-Catalyzed Degradation of Methanethiol in Humans

As mentioned above, exposure of humans to methanethiol is thought to occur mainly via the distal gastrointestinal tract due to the degradation of dietary methionine by gut microbiota. The rapid detoxification of methanethiol in the colonic mucosa through oxidative conversion to H_2_S and, further on, to thiosulfate has already been reported, more than 20 years ago [29,30], but the executing enzymes remained elusive at that time. In 2018, SELENBP1 was identified to act as a methanethiol oxidase (MTO), catalyzing the oxidation of methanethiol to formaldehyde, hydrogen peroxide (H_2_O_2_), and H_2_S in the presence of oxygen (Figure 1) [18]. Subsequently, H_2_S is metabolized to thiosulfate and sulfate in the sulfide oxidation unit, comprising four enzymes located in mitochondria [31].

We recently identified copper as a cofactor required for the MTO activity of SELENBP1, while selenium binding was dispensable in this regard [32]. Besides human SELENBP1, the orthologous proteins from the nematode *Caenorhabditis elegans* (SEMO-1) [32,33] and from the bacterium *Hyphomicrobium* sp. (SBP56) [34] have been shown to be copper-dependent MTOs. In fact, the amino acid sequences of SELENBP1 orthologs from different species contain histidine-rich metal-binding motifs [35]. SELENBP1 orthologs from mammals (*Homo sapiens*, *Rattus norvegicus*) [32,36], plants (*Arabidopsis thaliana*) [37], and bacteria (*Hyphomicrobium* sp.) [34] have been reported to bind or be associated with divalent cations such as copper, zinc, magnesium, and/or cadmium.

Even though SELENBP1 is ubiquitously expressed in human tissues, it is particularly abundant in the intestinal epithelium (https://www.proteinatlas.org/ENSG00000143416-SELENBP1/tissue, accessed on 5 July 2023), with a gradient in SELENBP1 expression along the crypt-luminal axis [38]. Immunohistochemical analysis of human colon biopsies from healthy donors revealed that SELENBP1 levels were highest in the epithelial cells located at the tip of the villi [38], which are in close contact with substances entering from the colonic lumen. Besides microbiota-derived methanethiol, SELENBP1 may also accept structurally related alkyl thiols as substrates, some of which arise from the digestion of dietary sulfur-containing phytochemicals [32].

Cellular differentiation appears to be a major trigger of SELENBP1 expression: both the spontaneous and butyrate-induced differentiation of proliferating Caco-2 cells, a human intestinal adenocarcinoma cell line, to a colonocyte-like phenotype is associated with induction of SELENBP1 gene and protein expression and an increase in MTO activity in the terminally differentiated cells [9,38,39]. The downregulation of SELENBP1 via the treatment of Caco-2 cells with small interfering RNA (siRNA) resulted in lowered expression of a differentiation marker of colonic epithelial cells, carcinoembryonic antigen [38]. SELENBP1 also became upregulated in the course of differentiation of HT29 cl.16E cells, an in vitro model of intestinal secretory cells, to a goblet-cell-like phenotype [38]. In vivo, an increase in SELENBP1 levels has been observed to occur along the colonic crypt–luminal axis, with the terminally differentiated epithelial cells at the top of the crypts showing the highest SELENBP1 expression [38]. Moreover, SELENBP1 has been reported to be a marker protein of terminally differentiated erythrocytes and adipocytes [40,41]. SELENBP1 is a highly abundant non-heme protein in erythrocytes [42], where it is presumably responsible for the degradation of circulating methanethiol. Indeed, MTO activity in whole blood has been attributed predominantly to the erythrocyte fraction [18]. Regarding adipocytes, SELENBP1 appears to be not only a marker but also an endogenous stimulator of terminal differentiation that is promoted by the signaling mediator H_2_S: the knockdown of SELENBP1 in murine 3T3-L1 pre-adipocytes, thus, resulted in the suppression of common features of adipocyte differentiation, such as intracellular lipid accumulation and the induction of adiponectin expression [43].

Two of the products of the SELENBP1-catalyzed oxidation of methanethiol, H_2_O_2_, and H_2_S, may exert pleiotropic effects and bivalent (both stimulatory and inhibitory) actions on signaling pathways, depending on their concentration and cellular localization as well as the (patho)physiological context. Both molecules are cytotoxic at high concentrations, whereas they are implicated as second messengers in redox signaling at lower concentrations. Thus, they may affect a wide variety of cellular processes, including differentiation, apoptosis, antioxidant adaptation, and mitochondrial respiration [44,45]. In addition, H_2_S may serve as an electron donor for the generation of ATP in the mitochondrial respiratory chain [46]. However, it still needs to be explored to what extent SELENBP1 contributes to the generation of H_2_O_2_ and H_2_S in different types of cells and whether biological effects of SELENBP1 other than methanethiol detoxification could be attributed to the MTO reaction products.

In addition to the intestine, several other tissues in the human body show high SELENBP1 levels, including the liver, the lungs, and the nasopharynx (https://www.proteinatlas.org/ENSG00000143416-SELENBP1/tissue, accessed on 5 July 2023). The liver appears to be involved in the detoxification of circulating methanethiol, as elevated concentrations of methanethiol were measured in the blood of patients suffering from liver cirrhosis with and without hepatic encephalopathy, and increasing methanethiol concentrations over time were associated with a progressive decline in the health status of the patients [47]. Also, the lungs might be capable of metabolizing methanethiol circulating in the blood, while methanethiol produced by bacteria populating the oral cavity might be detoxified in the mucosa of the nasopharynx. In this regard, elevated methanethiol concentrations have been measured in the oral cavity of patients with chronic periodontitis and intra-oral halitosis [48].

Interestingly, SELENBP1 was identified to act as MTO in a study based on five patients suffering from extra-oral halitosis, who were found to possess biallelic single-nucleotide polymorphisms (SNPs) in the *SELENBP1* gene. The patients showed up with cabbage-like malodor, due to high levels of methanethiol and another VSC, dimethyl sulfide (DMS), in their breath. The detected point mutations caused exchanges of single amino acids (Gly225Trp, His329Tyr) in SELENBP1 that resulted in a loss of MTO activity and, in turn, the accumulation of methanethiol and DMS in breath and body fluids. DMS is generated via the methylation of methanethiol under MTO-deficient conditions (Figure 1). The homologous overexpression of wild-type SELENBP1 restored the MTO activity in fibroblasts prepared from one of the patients [18].

## 4. Elevated Levels of Methanethiol and Its Derivatives in Various Types of Cancer

Both the generation and metabolism of methanethiol can be altered/dysregulated in cancer, resulting in elevated levels of methanethiol and methanethiol-derived metabolites in tumor tissue. Since methanethiol is highly volatile and membrane-permeant, the footprint of its dysregulated metabolism is found in both the body fluids, such as blood and urine, and excreted gases, such as breath and flatus, of cancer patients.

Differences in gut microbiota with respect to composition and relative abundance have been reported between healthy persons and patients with CRC [49]. Among the methanethiol-generating bacteria in the colon, *Fusobacterium nucleatum* was over-represented in CRC patients [50]. Correspondingly, higher methanethiol concentrations have been measured in the flatus of CRC patients, as compared to healthy individuals [13,51].

Elevated methanethiol concentrations were also found in the exhaled breath of persons diagnosed with oral squamous cell carcinoma [52], probably deriving from the high endogenous synthesis in the tumor tissue. The uptake of glucose and methionine is often elevated in cancer cells due to the upregulation of the respective transporters [5,6,7]. The accumulation of these two metabolites in cancer cells may favor a Maillard reaction between them, resulting in the non-enzymatic synthesis of methanethiol and its subsequent release, as demonstrated in in vitro experiments [51]. Moreover, methionine serves as a precursor for the biosynthesis of the methyl donor SAM that is catalyzed by the enzyme methionine adenosyl transferase 2A (MAT2A). Elevated MAT2A levels were observed in various types of cancer, including CRC, hepatocellular carcinoma (HCC), and breast and endometrial cancers, and are mostly considered as an unfavorable prognostic marker (https://www.proteinatlas.org/ENSG00000168906-MAT2A/pathology, accessed on 17 July 2023) [53]. As mentioned above, METTL7B may use SAM to produce methanethiol via the methylation of H_2_S [26]. Like MAT2A and the methionine transporter SLC43A2, METTL7B is upregulated in several types of cancer (https://www.proteinatlas.org/ENSG00000170439-METTL7B/pathology, accessed on 17 July 2023), thus providing a molecular rationale for elevated methionine-derived enzymatic methanethiol production in cancer cells. In thyroid cancer, where METTL7B is highly expressed, METTL7B has recently been shown to promote metastasis by increasing migration and invasion [54].

Another strong hint pointing to the dysregulation of methanethiol metabolism in cancer stems from observations that the methanethiol-degrading enzyme SELENBP1 is often and markedly downregulated in cancer tissue. Lowered expression of SELENBP1 has been reported for many types of cancer, such as tumors of the colon, lungs, ovaries, prostate, liver, thyroid, kidneys, and breast [19]. SELENBP1 has been designated a tumor suppressor that may inhibit cell proliferation, angiogenesis, metastasis, and resistance to chemotherapy, as well as promote apoptotic cell death (Figure 2) [19,55,56,57,58,59]. Moreover, low SELENBP1 levels in cancer tissue correlate with poor clinical prognosis for the patients [19].

MTO activities in tumor tissues of patients have not been assessed yet; however, severely decreased levels of an enzyme are usually associated with a diminished capacity to convert its substrate. In this regard, the erythrocytes and different tissues of SELENBP1-knockout (KO) mice were shown to be deficient in MTO activity and DMS, a biomarker of defective methanethiol oxidation accumulated in the plasma of the KO mice. Compared to the wild-type mice, even the heterozygous carriers exhibited less MTO activity in their erythrocytes and moderately elevated plasma levels of DMS [18]. It can be assumed that the catabolism of methanethiol in cancer patients is similarly shifted from predominant oxidation to methylation. Under the conditions of low SELENBP1 and high METTL7B expression in cancer cells, METTL7B is then likely to catalyze the methylation of a part of the excessive methanethiol to DMS. Indeed, DMS levels were reported to be elevated in the breath of patients with HCC and lung cancer, as compared to healthy controls [60,61]. High concentrations of DMS were also measured in the headspace of the tumor-derived human HepG2 hepatoma and A549 lung adenocarcinoma cell lines [62]. Some DMS may be oxidized to dimethyl sulfoxide (DMSO) and, further on, to dimethyl sulfone (DMSO_2_) (Figure 1) [18]. Elevated levels of DMSO_2_, the stable end product of this pathway, were detected in melanomas and endometrial and ovarian carcinomas [63,64,65]. Polysulfides, such as dimethyl trisulfide, are additional products of a dysregulated methanethiol metabolism and have been found among the volatiles emitted by melanoma cells [63]. Following subcutaneous injection of melanoma cells, both DMSO_2_ and dimethyl trisulfide concentrations were found to be increased in the urine of tumor-bearing mice, as compared to healthy controls [66].

Besides being excreted or methylated, excessive methanethiol may modify cysteine residues in proteins, as demonstrated for a methanethiol adduct of Cys34 in human albumin, which was more abundant in the serum of CRC patients compared to healthy persons [67].

Cancerous ulcers are probably the most suitable tumor tissue for the direct detection of characteristic volatiles in cancer emissions since their volatilome is not metabolized prior to release. Cancerous ulcers, or fungating cancer, occur when malignant tumor cells invade and erode the skin. This is often associated with the emission of unpleasant odors. The predominant odor-producing component in fungating head, neck, and breast cancers has been identified as the methanethiol-derived metabolite dimethyl trisulfide [68,69]. In this regard, it should be noted as well that ulcers are often infected, and a microbial origin of the detected dimethyl trisulfide cannot be ruled out.

## 5. Conclusions and Outlook: Methanethiol as a Promising Biomarker for Non-Invasive Cancer Diagnosis

Dysregulated metabolism in cancer cells results in the generation of cancer-associated volatile organic compounds (VOCs). Their measurement in body fluids, exhaled breath, and flatus may provide a tool for early non-invasive diagnosis of cancer. Among the identified cancer-associated VOCs are hydrocarbons, alcohols, aldehydes, ketones, and VSCs, such as H_2_S and methanethiol [11,12,13,70,71,72]. Indeed, artificial intelligence (AI)-based surface-enhanced Raman spectroscopy (SERS) has recently been demonstrated as a useful and accurate method for the early diagnosis of oral cancer through the detection of methanethiol in exhaled breath [14]. Elevated concentrations of methanethiol in cancer patients may derive from increased exogenous and endogenous synthesis and suppressed degradation. Moreover, the downregulation of the methanethiol-oxidizing enzyme SELENBP1 that is observed in many types of cancer may shift its metabolism to the generation of methylated derivatives, which have been detected in cancer patients as well (Figure 3).

In this regard, the use of canine olfaction for non-invasive cancer diagnosis is also of high interest. Dogs have an extraordinary sense of smell with a detection limit of 1 ppt, exceeding by far the capability of humans [73,74]. They are able to identify cancers of the lung, bladder, breast, colon, ovary, prostate, and skin by sniffing the breath, urine, flatus, tumor tissue, or even cultured cancer cells [75,76,77,78,79]. The olfactory profile that is sniffed by trained dogs may, in part, result from dysregulated methanethiol metabolism in cancer patients. In a comparative analysis of volatiles in the exhaled breath of lung cancer patients employing solid-phase microextraction/gas chromatography–mass spectrometry (SPME/GC-MS) and subsequent canine detection, methanethiol-derived dimethyl disulfide was found to be the main compound responsible for discriminating between healthy individuals and cancer patients [80]. Moreover, volatile nitrosothiols that may result from the reaction of alkyl thiols with nitric oxide (NO) were reported to be recognized by a sniffer dog [81].

Taken together, elevated concentrations of methanethiol and its methylated derivatives in body fluids and excreted gases contribute to the “scent of cancer”, frequently observed in affected patients.

## Figures and Tables

**Figure 1 antioxidants-12-01780-f001:**
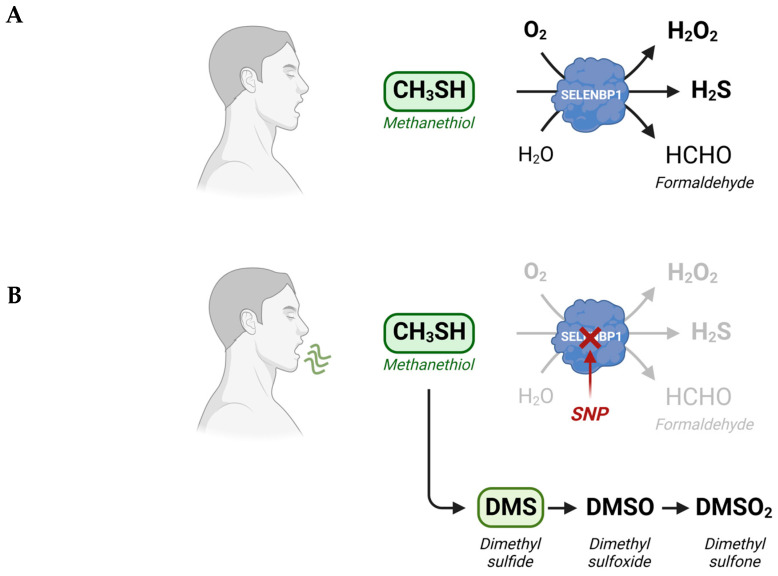
Alternate metabolic fates of methanethiol. (**A**) Physiologically, degradation of methanethiol occurs predominantly through SELENBP1, which acts as MTO, catalyzing the rapid oxidative conversion of methanethiol to H_2_S, H_2_O_2_, and formaldehyde. (**B**) In SELENBP1-deficient cancer cells and in cells that possess an inactive MTO due to mutations (SNPs) in the *SELENBP1* gene, methanethiol is methylated to DMS, which can be oxidized subsequently to DMSO and DMSO_2_. Methanethiol and DMS may be detected in the exhaled breath of afflicted persons. Scheme created with Biorender.com.

**Figure 2 antioxidants-12-01780-f002:**
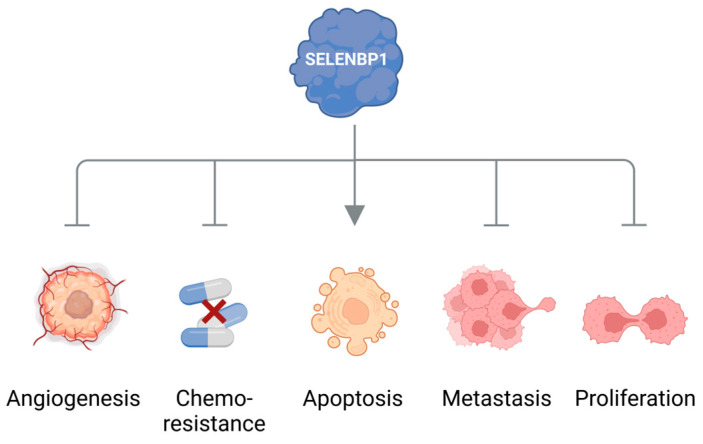
SELENBP1 as a tumor suppressor. SELENBP1 has been reported to inhibit cell proliferation, angiogenesis, metastasis, and resistance to chemotherapy as well as to promote apoptotic cell death. Scheme created with Biorender.com.

**Figure 3 antioxidants-12-01780-f003:**
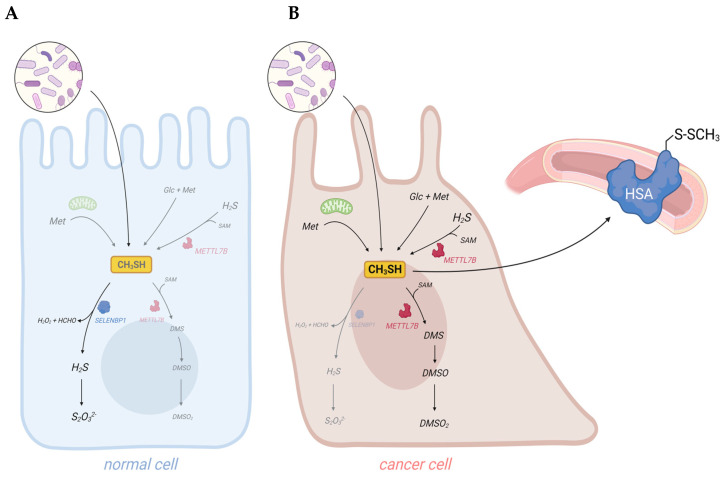
Dysregulation of sulfur metabolism in cancer cells may result in elevated levels of methanethiol and its derivatives in body fluids and excreted gases of cancer patients. (**A**) In healthy persons, the vast majority of methanethiol derives from the metabolic activity of gut bacteria. Some methanethiol may diffuse into the colonic epithelium, where it is rapidly oxidized by SELENBP1. (**B**) In cancer patients, endogenous synthesis of methanethiol may substantially increase due to elevated levels of metabolites, such as methionine (Met), glucose (Glc), H_2_S, and SAM, in tumor cells. Downregulation of SELENBP1 and upregulation of METTL7B may result in a shift from oxidation to methylation of methanethiol. As methanethiol and its methylated derivatives are highly volatile, they may be used as biomarkers for non-invasive cancer detection. Adducts of methanethiol with proteins may also be detected, as it has been demonstrated for human serum albumin (HSA). Scheme created with Biorender.com.

## Data Availability

Not applicable.

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
