# Peer review of "Methanethiol: A Scent Mark of Dysregulated Sulfur Metabolism in Cancer"

_antioxidants, 2023, doi:10.3390/antiox12091780_

Round 1

Reviewer 1 Report

Methanethiol is a volatile sulfur compound (VSC) produced by gut bacteria from dietary methionine or endogenous enzymatic and non-enzymatic reactions in cancer cells. Methanethiol contributes to the characteristic odor of cancer and may serve as a biomarker for non-invasive diagnosis. SELENBP1 is an enzyme that catalyzes the oxidation of methanethiol to H2S, H2O2, and formaldehyde. SELENBP1 is highly expressed in the intestinal epithelium and other tissues involved in methanethiol detoxification. SELENBP1 is considered a tumor suppressor that inhibits various aspects of cancer progression. Many types of cancer show downregulation of SELENBP1 and upregulation of methanethiol-producing enzymes such as METTL7B. This results in elevated levels of methanethiol and its methylated derivatives, such as DMS, DMSO, and DMSO2, in tumor tissue, body fluids, and exhaled breath of cancer patients. Specific comments:

1.          The title of the paper is not very informative and does not reflect the main focus of the review, which is the dysregulation of methanethiol metabolism in cancer and its potential as a biomarker. For example, a more specific title could be “Methanethiol: A scent mark of dysregulated sulfur metabolism in cancer”.

2.          A concise abstract should summarize the review’s main points, such as the sources and pathways of methanethiol production and degradation in humans, the alterations of methanethiol metabolism in cancer, and the prospects of methanethiol as a non-invasive cancer diagnostic tool.

3.          The introduction should provide more background information on the role of sulfur metabolism in cancer and the current challenges and limitations of existing cancer biomarkers. The authors should also state the aim and scope of their review and highlight the main questions or hypotheses they intend to address.

4.          The section on exogenous and endogenous methanethiol production in humans is well-written and informative. Still, the authors should also discuss the factors that influence the inter-individual variability of methanethiol production, such as diet, genetics, microbiota composition, and environmental exposure.

5.          The section on SELENBP1-catalyzed degradation of methanethiol in humans is also comprehensive and clear. Still, it could benefit from more references to support some of the statements, such as the role of copper as a cofactor for SELENBP1, the effects of H2O2 and H2S as signaling molecules, and the association of SELENBP1 expression with cellular differentiation.

Reviewer 2 Report

The manuscript with the title Methanethiol- a scent mark of cancer reveals the pathways in which microbiota interfers in the degradation of volatile organic componds.It is particulair interesting as it focuses on cancer non invasive diagnostic, while methanethiol is an important player. The review is well written, existing literature is well represented, the authors made it easy to read and understand for general public, not only for specialists.The fact that the review is short is a plus.

The introduction covers in my opinion the subject very well. Being a short review it adds a plus. The fact that some basic info are presented and explained in figures makes it more readable by all the specialists, not only gastroenterologists or oncologists. 

Microbiota is a very delicate matter and the authors describe the bacteria involved.

Novelty is presented. Conclusions are in line with the findings.

I find the need to recommend one suggestion, a list of abreviations, this would make th e review easier to follow.

I recommend for publication.

Round 2

Reviewer 1 Report

Methanethiol is a volatile sulfur compound (VSC) produced by gut bacteria from dietary methionine or endogenous enzymatic and non-enzymatic reactions in cancer cells. Methanethiol contributes to the characteristic odor of cancer and may serve as a biomarker for non-invasive diagnosis. SELENBP1 is an enzyme that catalyzes the oxidation of methanethiol to H2S, H2O2, and formaldehyde. SELENBP1 is highly expressed in the intestinal epithelium and other tissues involved in methanethiol detoxification. SELENBP1 is considered a tumor suppressor that inhibits various aspects of cancer progression. Many types of cancer show downregulation of SELENBP1 and upregulation of methanethiol-producing enzymes such as METTL7B. This results in elevated levels of methanethiol and its methylated derivatives, such as DMS, DMSO, and DMSO2, in tumor tissue, body fluids, and exhaled breath of cancer patients. The revision of the manuscript is much improved, no additional comments.